Communication

# Effectiveness of different recruitment strategies in an RCT of a surgical device: experience from the Endobarrier trial

Aruchuna Ruban ![ORCID],[1] Christina Gabriele Prechtl,[2] Michael Alan Glaysher,[3] Navpreet Chhina,[4] Werd Al-Najim,[5] Alexander Dimitri Miras,[6] Claire Smith,[7] Anthony P Goldstone,[4] Mayank Patel,[8] Michael Moore,[9] Hutan Ashrafian ![ORCID],[1] James P Byrne,[10] Julian P Teare[1]

AR and CGP are shared first authors.

For numbered affiliations see end of article.

**Correspondence to**
Dr Aruchuna Ruban;
aruchuna@doctors.org.uk

## ABSTRACT

Recruiting participants into clinical trials is notoriously difficult and poses the greatest challenge when planning any investigative study. Poor recruitment may not only have financial ramifications owing to increased time and resources being spent but could adversely influence the clinical impact of a study if it becomes underpowered. Herein, we present our own experience of recruiting into a nationally funded, multicentre, randomised controlled trial (RCT) of the Endobarrier versus standard medical therapy in obese patients with type 2 diabetes. Despite these both being highly prevalent conditions, there were considerable barriers to the effectiveness of different recruitment strategies across each study site. Although recruitment from primary care proved extremely successful at one study site, this largely failed at another site prompting the implementation of multimodal recruitment strategies including a successful media campaign to ensure sufficient participants were enrolled and the study was adequately powered. From this experience, we propose where appropriate the early engagement and investment in media campaigns to enhance recruitment into clinical trials. Trial Registration: ISRCTN30845205.

## INTRODUCTION

Obtaining a satisfactory outcome in any clinical trial is largely underpinned by a successful recruitment campaign to drive participant's numbers and to ensure that the study is adequately powered for the results obtained. The recruitment process poses the greatest challenge for those involved in conducting clinical trials.[1] Attempts to negate poor recruitment can include lengthening the recruitment timeline or broadening the screening criteria, which can have a detrimental impact on the cost of the trial or indeed dampen the clinical effect of a particular intervention. Ultimately, if recruitment goals are not reached, this can potentially lead to the early termination of a trial. A review of the National Cancer Institute Therapy Evaluation Programme (CTEP) sponsored oncology trials found that 38%

failed to attain the minimum accrual goals with 71% of phase III trials resulting in poor accruals.[2] A positive correlation was found between poor accrual rates and longer development time of clinical trials—the time from initial concept to commencement of the trial.

Often the time taken to recruit patients to a clinical trial is grossly underestimated. A study of 20 multicentre national randomised controlled trials (RCTs) funded by the National Health and Medical Research Council (NHMRC) found that the average recruitment period was 4–5 years, excluding the period required for participant follow-up.[3] A recently published multicentre diabetes prevention trial of men took 7 years to recruit the study screening over 19 000 participants in order to enrol 1007 (5%) participants.[4]

There are ethical implications associated with early trial termination due to inadequate participant recruitment. First, patients already recruited into the study may be exposed to potentially harmful interventions despite the outcome of the trial being fruitless. Second, an early terminated clinical trial will invariably lead to delays in a new treatment or drug therapy being made commercially available as outstanding questions may still remain on its efficacy or safety profile. Failed clinical trials not only waste resources and funding but also the time of patients and researchers.

Research funding bodies will expect to see evidence of meticulously planned recruitment strategies to ensure that any grants approved are utilised appropriately and that sufficient participant numbers are obtained for a trial in order to address the primary research question.

In this article, we reflect on our own personal experience from recruiting to a nationally funded multicentred RCT designed to investigate and compare the

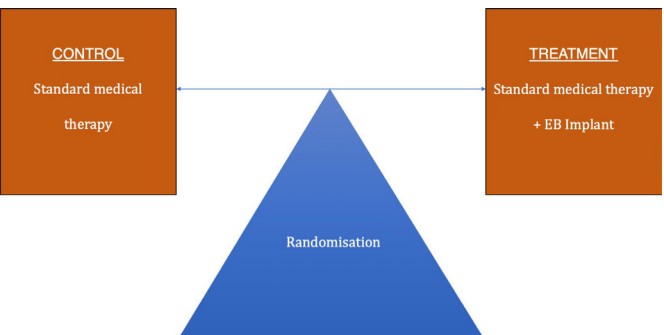

**Figure 1** Randomisation. EB, Endobarrier.

effect of the Endobarrier duodenal jejunal bypass liner with standard medical therapy in the treatment of obese patients with type 2 diabetes mellitus (T2DM).[5] This trial commenced in 2014 at Imperial College Healthcare NHS Trust, London, and University Hospital Southampton (UHS) NHS Foundation Trust, UK, and concluded in January 2019.

### The Endobarrier trial

The Endobarrier RCT is funded by the National Institute for Health Research (NIHR) and is part of the Efficacy and Mechanism Evaluation programme (EME). EME is a partnership between the Medical Research Council (MRC) and NIHR and was primarily set up to support clinical trials that test the efficacy of interventions. The study was conducted at two investigational sites in the UK, Imperial College Healthcare NHS Trust (ICHT), which includes St Mary's Hospital and Hammersmith Hospital, and UHS NHS Foundation Trust. This was a 2-year study in which 170 eligible patients with obesity and T2DM were recruited and randomised to either the control or the treatment arm group (figure 1).

The treatment arm received the Endobarrier device for 1 year in addition to standard medical therapy and was followed up for a further 1 year. The control group received standard medical therapy and lifestyle intervention therapy alone over the period of 2 years. The trial protocol including all details on the Endobarrier device and trial design has been previously published by our group.[5]

### METHODS OF RECRUITMENT

The target population for this study were men and women, aged 18–65 years, and obese (BMI >30 kg/m$^2$) with T2DM but adequate insulin reserve. The study eligibility criteria are shown in box 1.

Three of the eligibility criteria (identified by an asterisk) were modified from the original protocol to broaden the eligibility criteria in order to recruit more participants:

1. HbA1c upper limit was extended to 97 mmol/mol from 86 mmol/mol.
2. Criterion for liver and kidney disease was modified from 'Severe liver (AST, ALT or gGT >4 times upper limit) or kidney failure (serum creatinine >180 mmol/L), es-

## Box 1 Study eligibility criteria

**Inclusion criteria**
1. Age 18–65 years (men or women).
2. Type 2 diabetes mellitus for at least 1 year.
3. HbA1c 7.7%–11.0% equivalent to 58–97 mmol/mol*.
4. On oral hypoglycaemic medications.
5. BMI 30–50 kg/m$^2$.

**Exclusion criteria**
1. Language barrier, mental incapacity, unwillingness or inability to understand and be able to complete questionnaires.
2. Non-compliance with eligibility criteria.
3. Women of childbearing potential who are pregnant, breast feeding or intend to become pregnant or are not using adequate or reliable contraceptive methods.
4. Evidence of absolute insulin deficiency as indicated by clinical assessment, a long duration of T2DM and a fasting plasma C-peptide of <333 pmol/L.
5. Current use of insulin.
6. Previous diagnosis with type 1 diabetes mellitus or a history of ketoacidosis.
7. Requirement of non-steroidal anti-inflammatory drugs or prescription of anticoagulation therapy during the implant period.
8. Current iron deficiency and/or iron deficiency anaemia.*
9. Symptomatic gallstones or kidney stones at the time of screening.
10. History of coagulopathy, upper gastrointestinal bleeding conditions such as oesophageal or gastric varices, congenital or acquired intestinal telangiectasia.
11. Previous gastrointestinal surgery that could affect the ability to place the device or the function of the implant.
12. History or presence of active *Helicobacter pylori* (if subjects are randomised into the EndoBarrier arm and have a history or presence of active *H. pylori* tested at study visit 2 they can receive appropriate treatment and then subsequently enrol into the study).
13. Family history of a known diagnosis or pre-existing symptoms of systemic lupus erythematosus, scleroderma or other autoimmune connective tissue disorder.
14. Severe liver impairment (ie, AST, ALT or gGT >4 times upper limit of the reference range) or kidney impairment (ie, estimated glomerular filtration rate (GFR) <45 mL/min/1.73 m$^2$. *
15. Severe depression, unstable emotional or psychological characteristics (including Beck Depression Inventory II score >28).
16. Poor dentition and inability to adequately chew food.
17. Planned holidays up to 3 months following the EndoBarrier Implant.

timated glomerular filtration rate (GFR) cut-off is 60' to 'Severe liver (AST, ALT or gGT >4 times upper limit) or kidney impairment estimated glomerular filtration rate (GFR) <45 mL/min/1.73 m$^2$'.

3. 'History of iron deficiency/iron deficiency anaemia' was modified to be more specific to 'current iron deficiency or iron deficiency anaemia'.

As the vast majority of patients with T2DM are managed in the primary care setting, it was anticipated that general practices would provide the most valuable resource in which to identify eligible patients and this was supported by initial analysis of Diabetes Research Network (DRN) databases. There are eight Diabetes Research Network (DRN) hubs nationally with one covering ICHT and a second hub covering UHS. In North West London, there

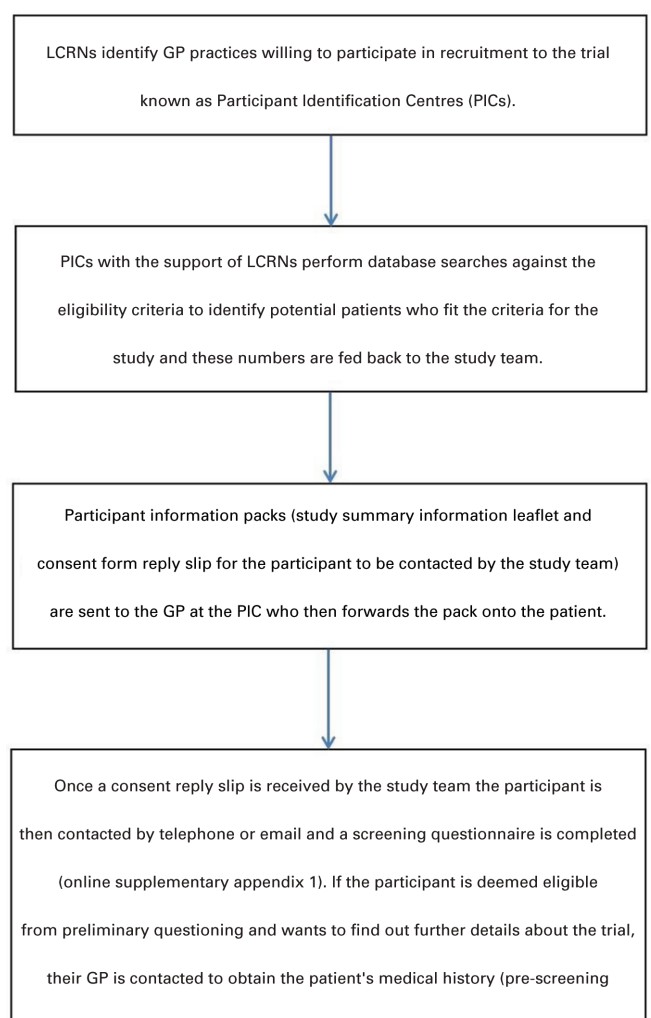

**Figure 2** Participant recruitment from GP PICs. LCRN, Local Clinical Research Network.

were 416 general practices (GP) with a total list size of 2 148 746 and the number of adults registered with T2DM was 98 842. When planning this study, the database of two such GP in London was interrogated and within these two practices alone over 500 patients were identified who matched the age profile, HbA1c and BMI criteria for this study. In Southampton, experiences from a previously similar NIHR-funded diabetes study that had recruited successfully from primary care suggested that a sample of 10–20 practices would be adequate to identify sufficient numbers for this study. Based on this preliminary analysis, we were confident that participants fitting the criteria for the study could be fairly easily identified from the primary care databases.

GP practices in the region were approached in the first instance by the Local Clinical Research Network (LCRN) on behalf of the study team. The LCRNs are an initiative set up by the NIHR to coordinate and support the delivery of research across the NHS in England. They fund teams of research staff to enter hospitals and GP in

order to facilitate and increase awareness of the research opportunities available to patients.

Since 2009, the North West London (NWL)-DLRN had established a hub and spoke model to support diabetes research in primary care, which currently provided access to over 13 000 people with T2DM. This model was available to study teams working on diabetes-related NIHR portfolio studies. The process of recruitment from GP is summarised in the flow diagram (figure 2).

This process was reimbursed by the LCRN with £150 paid for the GP database search and set up, £0.60 per participant information pack sent to patients and £40 for each GP pre-screening questionnaire completed. Once a GP practice agreed to act as Patient Identification Centre (PIC) for our trial, they initiated database searches to identify potential participants using two of the main inclusion criteria (BMI $>30 \, \text{kg/m}^2$ and diagnosis of T2DM). The final number of eligible patients was then communicated back to the LCRN or research site who populated the adequate amount of patient packs (including patient information summary leaflet, recruitment invitation letter with response slip and prepaid envelope) and posted them back to the GP. The GP sent the packs out to each identified patient from their database. The same method was used across the two research sites, Imperial College Healthcare NHS Trust and UHS NHS Trust.

Various other strategies were also employed to compliment recruitment from GP practices. These included:
1. Diabetes Alliance for Research England (DARE) registry—a database of 60 000 patients nationwide with diabetes who have expressed an interest to be informed and participate in diabetes research. This database was interrogated and patients who met the criteria were sent out participant packs with information about the study.
2. Study website—official websites for the trial were set up at each research site through the media office at Imperial College London and University of Southampton (https://www.imperial.nhs.uk/research/research-trials/diabetes-research-trials; https://www.southampton.ac.uk/medicine/academic_units/projects/endobarrier.page) and by the Imperial College research facility: http://imperial.crf.nihr.ac.uk/studies/endobarrier/
3. Diabetes UK—contacted charities including diabetes UK who promoted the study on their website and magazine.
4. Social media—Facebook posts and Twitter feeds.
5. Posters and leaflets—were placed in prominent areas in GP practices, diabetes and renal outpatient clinics.
6. Newspaper advertising—weekly adverts were placed in local newspapers in London (*The Evening Standard* and *Metro*) and in Southampton (*Bournemouth Echo, Daily Echo, The News*) over different time periods.

The imperial clinical trials unit received regular updates on recruitment numbers and sources from each research site. This helped identifying recruitment challenges early

**Table 1** Key dates in recruitment process

| Key events | Imperial College Healthcare | Southampton Hospital |
|---|---|---|
| Research site initiation | 20 October 2014 | 30 April 2015 |
| First participant screened | 28 November 2014 | 3 July 2015 |
| First participant randomised | 6 March 2015 | 9 July 2015 |
| Final participant randomised | 18 October 2016 | 14 October 2016 |

**Table 2** Sources of recruitment

| Patient sources of recruitment | Imperial college healthcare NHS trust | University hospital southampton | Total |
|---|---|---|---|
| GP | 65 | 397 | 462 |
| Newspaper adverts | 1004 | 102 | 1106 |
| Study website | 75 | 9 | 84 |
| DARE | 16 | 0 | 16 |
| Other bariatrics and diabetes clinics | 9 | 9 | 18 |
| Diabetes UK | 7 | 16 | 23 |
| Other: research/science museum | 7 | 0 | 7 |
| Poster | 4 | 3 | 7 |
| Telescreen outpatient clinics | 4 | 0 | 4 |
| Radio station interview | 0 | 2 | 2 |
| Social media (Facebook or Twitter) | 4 | 0 | 4 |
| Friend | 1 | 1 | 2 |
| Other/unknown | 14 | 28 | 42 |
| Total | 1210 | 567 | 1777 |

Recruitment at Imperial College London Healthcare NHS Trust (ICHT).

and enabled the research teams to put new recruitment sources in place where necessary.

## RESULTS

The key dates for the recruitment process are outlined in table 1, and the progress of recruitment from commencement through to completion in mid-October 2016 is shown in figure 3.

Recruitment was initially anticipated to take only 12 months, but in light of the early termination of the pivotal ENDO trial in the USA there were significant delays to the final ethical and local approvals being granted. As such, the recruitment period was extended to 24 months. The first 18 months focused on identifying PICs from the primary and secondary care setting. Presentations were made at local GP practices, meetings were held with nurse specialists and endocrinologists working in the community diabetes practices. Press releases were made online, on social media sites and in major tabloid newspapers. The recruitment outcomes from these different sources are summarised in table 2. A flowchart summarising the overall recruitment figures from initial participant contact right through to randomisation is depicted in figure 4. More details of the process of telephone screening (online supplementary appendix 1), screening visit and randomisation can be found in the previously published protocol paper for this trial.[5]

ICHT is located in North West London, which encompasses a population of around 2.4 million people and it is estimated that around 40% of GP practices in the region are engaged and recruiting into clinical trials. The North West London LCRN provided the link between the study team and these local GP practices and supported recruitment in this region.

Unfortunately, despite these links, only 65 responses were received from patients via GP PICs. As such, recruitment strategies were modified to focus on media platforms and, as a consequence, the majority of patients recruited at the ICHT site were self-referred after hearing about the study from newspapers adverts. Over 1000 phone calls were received from patients following the newspaper adverts.

From November 2015 through to September 2016, a quarter page advert was placed in two London newspapers—the *Metro* and *Evening Standard*. This included a digital advertising campaign in which adverts were also placed within the desktop and tablet versions of the newspaper, which provided a direct link to the trial study website when the advert was clicked on.

Table 3 is the activity summary data provided by the advertising company and includes the number of times the adverts were accessed online and the number of 'clicks', which refers to the number of people who clicked on the advert to directly access the study website. The Industry standard for the click through rate (CTR) is approximately 0.3% as provided by the advertising team from the *Metro* and *Evening Standard*.

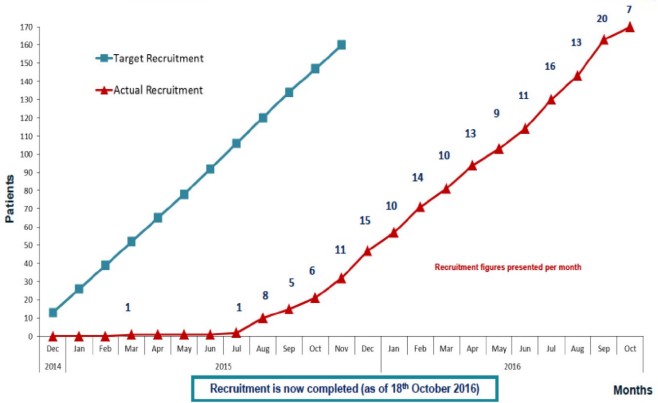

**Figure 3** Recruitment timeline.

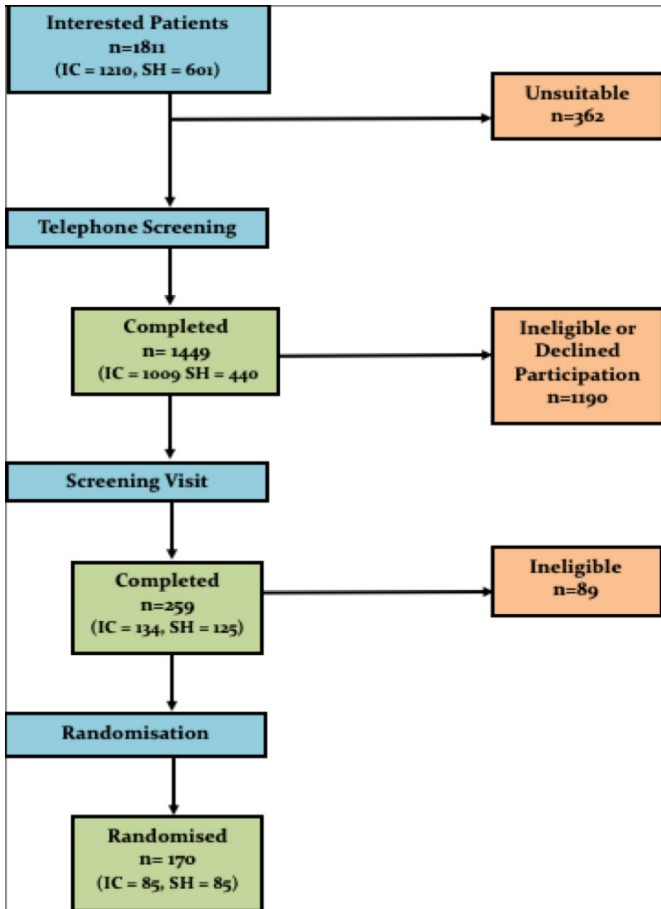

**Figure 4** Recruitment flow chart: SH=Research Site Southampton, IC=Research Site Imperial College.

### Recruitment at UHS

Research delivery in this region is supported by Wessex LCRN, which is inclusive of over 80 GP practices that are intimately involved in study recruitment. Unlike the experience of ICHT, GP PICs were by far the greatest source of eligible participants during recruitment at UHS. In total, UHS received 397 responses from patients identified via GP PICs, which was over six times more than the London site. However, despite this, the number of patients randomised from this cohort was still insufficient to reach the recruitment target. As such, a smaller newspaper advert campaign was launched in one local paper in Southampton in which 13 adverts were published during

June and July 2016. This generated 102 new telephone consults from patients (10 times less than the London site) and represented the second most successful source of randomised patients at this site.

### DISCUSSION
#### Difficulties in recruitment

Despite a clear strategy from the offset and taking into account a non-linear recruitment rate with a delayed start at the beginning of the trial, recruitment took much longer than anticipated taking 2 years to complete rather than initially predicted 1 year. As a result, an application had to be made to the NIHR (funding body) for a 1-year extension to the trial and, in addition, to request appropriate funding to support these extended activities. There are various explanations for the slow recruitment and poor response seen, and these are discussed below.

#### Variable uptake from general practice

Participant recruitment from primary care at the ICHT site was extremely disappointing, despite the initial forecast that the vast majority of trial participants would be recruited from primary care with support from the diabetes and primary care research networks.

More than 400 GP practices were approached but fewer than 10% of these agreed to become PICs and completed database searches on behalf of the study team. The workloads of primary care physicians are very high, and some may feel that it is not feasible to dedicate any further time to research as this might be at the detriment of their clinical practice. Similar disengagement from research by primary care practitioners has also been reported in a palliative care study.[6] In addition, GP practices in North West London may be saturated with calls for participation in clinical trials in the local area, as there are hundreds of clinical trials being conducted in the local region.

Database searches from agreed PICs revealed approximately 1200 patients as being suitable for the trial when matched against the eligibility criteria and participant packs were sent out to these patients. However, only 65 (5%) participants' reply slips were received. The small number of responses received lead to only 12 patients being invited for screening following their initial telephone consultation, of which 6 participants were

**Table 3** Summary of activity from digital adverts in the *Metro*

|  |  | Total page views | Clicks | CTR (%) |
|---|---|---|---|---|
| *Metro* | Digital newspaper advert* | 150 535 | 742 | 0.49 |
|  | Tablet advert* | 39 873 | 465 | 1.17 |
|  | Tablet advert† | 13 655 | 134 | 0.98 |
| *Evening Standard* | Digital newspaper advert | 150 396 | 180 | 0.52 |

*22–24 March 2017.
†21 April and the evening standard (19 and 21 April).
CTR, click through rate.

randomised into the trial. This is in stark contrast to the UHS site where 397 reply slips were received in response to a similar number of participant packs being sent out.

## Variable uptake from trial participants

There are a few potential explanations for why many patients declined to contact the study team:

1. Participant information sheet (PIS) suboptimal: The PIS may have contained too much information or may have made the study sound overcomplicated or invasive, thus discouraging the participant from taking part. The PIS and participant reply slip were only available in English language, and some patients may not have been literate in English to understand and act on the information. Patient Public Involvement during the design stages was minimal and a possible reason for lower participant response rates. It is also important to note that there have been ethnical differences in the population that was approached at each research site with North West London representing a large Asian population and the area in and around Southampton representing a large white British population. However, a falls prevention RCT by Cockayne et al[7] failed to demonstrate any significant increase in participant recruitment or retention through the use of an optimised PIS.

2. The participant invitation letters may not have reached their intended recipient: According to figures from 2016/2017 published by the Ministry of Housing Communities and Local Government English Housing Survey Report, 30% of households in London are private renting with a further 22% renting in the social sector.[8] It is estimated that around 37% of private renters have moved three or more times in the last 5 years. Consequently, there is more chance of these letters being sent to the wrong address and not reaching the participant at all.

3. Saturation from clinical trial invites: Patients living in the London area may well receive multiple invites to participate in clinical trials so may chose to ignore these invites if they receive too many.

## ENDO trial suspension

The ENDO trial was a pivotal multicentred double-blinded RCT in the USA where subjects were randomised to either receive the device or sham treatment in order to assess the efficacy and safety of the device. The study opened in November 2012 but was terminated early by the US Food and Drug Administration in March 2015 after the development of 7 liver abscesses in 217 patients enrolled in the trial (3.2%). All patients with this complication were treated with antibiotics and, if necessary, draining with no permanent sequelae. The ENDO trial suspension had a direct impact on our study leading to a 3-month hiatus in our recruitment (from April to June 2015) as a substantial amendment to the study protocol and PIS was required to include the risk of hepatic abscess, which was quoted as 1%. This also meant that patients already recruited

to the trial had to be reconsented on their next visit to ensure they were aware of the potentially increased risk of hepatic abscesses.

## Lack of support staff

The newspaper advertising campaign was hugely successful generating numerous telephone calls and emails requiring urgent attention. On the days when the adverts featured in the newspapers, on average 30–50 telephone calls and emails were received by the study team at ICHT. Unfortunately, the infrastructure was not in place to deal with this unprecedented demand, which meant that not all telephone calls and emails were responded to promptly.

## Strict eligibility criteria

The eligibility criteria for the study were very stringent in order to ensure participant safety and to establish the appropriate diabetes status of those patients entering the study. Modifications to the eligibility criteria, which included raising the HbA1c range and lowering the kidney function (estimated GFR) cut-off, helped to widen the recruitment net.

## Reflecting on the success of newspaper advertising campaign

The fantastic response from the newspaper advertising campaign came as a surprise as reports in the literature are conflicting when judging the success of newspaper adverts for clinical trial recruitment, particularly when considering the high cost implications associated with such media campaigns.

The Scotland Standard Care vs Celecoxib Outcome Trial (SCOT) clinical trial investigating cardiovascular safety of non-steroidal anti-inflammatory drugs in patients with rheumatoid arthritis and osteoarthritis found little impact when they deployed a newspaper advertising campaign.[9] The study found that the adverts attracted relatively small numbers of respondents, and of those respondents most were not eligible to take part. This was in stark contrast to our adverts that generated a large number of respondents, from which we were able to recruit the vast majority of participant to the ICHT site.

An RCT conducted in Australia of vitamin E in the prevention of cataract and age-related maculopathy used five recruitment methods: newspaper advertising, radio advertising, GP practices, community groups and electoral roll mail-out.[10] Recruitment was successfully completed in the anticipated time frame with newspaper adverts and electoral roll mail-out found to be the most effective methods of participant recruitment in terms of both the absolute number of participant recruited and the cost per participant. Similar to our experience, the newspaper adverts generated a great deal of interest and a number of telephone calls, which placed a huge strain on the study team to respond to each inquiry in a timely fashion. In addition, they found that direct approaches to community groups or GP practices were not fruitful with the authors concluding that strong collaborative links

with GP practices may be necessary for this approach to be successful.

A similar study design to the Endobarrier trial was observed in a prospective multicentred RCT investigating Roux-en-Y Gastric Bypass (RYGB) versus intensive medical management for treatment of T2DM conducted at three institutions in the USA and one in Taiwan.[11] This trial successfully recruited 120 participants but this took 4 years and also involved lowering their BMI criteria and the addition of another centre to recruit more patients into the study. Two recruitment sites also used a mass media campaign and of the 120 randomised participants, 10% were recruited directly from newspaper adverts and 19% were from radio advertisements. The authors concluded that their recruitment could have been accelerated by enrolling more sites and by increasing the advertising budget.

The major benefit of using newspaper advertising is that it relies on a degree of self-motivation from the potential participant to contact the study team but also gets the message across in a non-intrusive way, as the advert is subtly placed in their daily newspaper hopefully sparking interest in the reader. Patients who contacted us appeared very keen to find out more information on the trial and were genuinely disappointed if they did not meet the study eligibility criteria.

One of the pitfalls encountered with the newspaper advertising is that only a small amount of information on the trial can be published in an advert, which meant more time spent on telephone calls to patients explaining more details of the study. One potential solution to this is to send a link to the study website in an automated email explaining where further information can be easily accessed. Newspaper advertising is also hugely expensive so can be a disaster if ineffective; the total cost across both our research sites to fund our adverts was £48 179.

It must also be noted that although recruitment from GP practices was poor at the London site, the same was not observed at our Southampton site where recruitment from primary care was considerably better. This is in line with a trial that recruited participants for physical activity for individuals with diabetes.[12] Researchers found that traditional recruitment approaches such as posting flyers and using clinical referrals were not successful whereby 77% of the participants were recruited using the electronic medical record system. This suggests that discrepancies in recruitment success in our trial could be site specific owing to the difference in patient populations between these two cities as previously identified.

## CONCLUSION

From our own experience, we strongly feel that at the planning stage of any clinical trial due consideration is given to media and advertising when the study design allows recruitment using this modality. Funding for future grant applications should be costed accordingly so that more resources can be devoted to newspaper adverts and social media campaigns. Equally having a dedicated study team to deal with the influx of calls and emails that might be generated through an advertising campaign is imperative so that responses occur swiftly and potential opportunities to recruit participants are not missed. Such team would ideally be headed by a clinician complimented by a research nurse and administrator.

It is clear that fundamental to any successful clinical trial is a successful recruitment campaign; obtaining the full quota of participants within a suitable time frame while using cost-effective methods. What is not so apparent is the best strategy to achieve this goal and so it is vital that there is flexibility in implementing variable recruitment modalities for multicentre trials across different regions in England and the rest of the UK.

**Author affiliations**
[1]Department of Surgery and Cancer, Imperial College London, London, UK
[2]Department of Public Health, Imperial College Healthcare NHS Trust, London, UK
[3]Division of Surgery, Southampton Biomedical Research Centre, University Hospital Southampton, Southampton, UK
[4]PsychoNeuroEndocrinology Research Group, Neuropsychopharmacology Unit, Centre for Psychiatry and Computational, Cognitive and Clinical Neuroimaging Laboratory, Division of Brain Sciences, Imperial College London, London, UK
[5]Dietician, Imperial College London, London, UK
[6]Division of Diabetes, Endocrinology and Metabolic Medicine, Imperial College Healthcare NHS Trust, London, UK
[7]Imperial Clinical Trials Unit, Imperial College London, London, UK
[8]Department of Diabetes and Endocrinology, University Hospital Southampton, Southampton, UK
[9]Primary Care Medical Group, University of Southampton Medical School, Southampton, UK
[10]Division of Surgery, University Hospital Southampton NHS Foundation Trust, University Hospital Southampton, Southampton, UK

**Acknowledgements** Special thanks are given to Emanuela Falaschetti and Nicholas Johnson who are dedicated statisticians for this clinical trial.

**Contributors** AR is a coinvestigator at Imperial College London, planned and designed the concept of the paper and is the corresponding and primary author of manuscript. CGP is a trial manager at Imperial College London and Imperial Clinical Trials Unit, co-author of this manuscript and contributed to the design and concept of this paper and approved the final version. MAG is a coinvestigator at University Hospital Southampton, contributed to the writing of the manuscript and approved the final version. NC is a coinvestigator at Imperial College London, contributed to the acquisition of the data and provided critical appraisal of current manuscript and approved the final version. WA-N is a coinvestigator at Imperial College London, contributed to the acquisition of the data and in writing the manuscript and approved the final version. ADM is a coinvestigator at Imperial College London, coauthor of recruitment methodology section, contributed to writing the manuscript and approved the final version. CS interpreted the data and provided critical appraisal of current manuscript and approved the final version. AG is a coinvestigator at Imperial College London and trial coapplicant, contributed to the writing of the manuscript and approved the final version. MP is a coinvestigator at University Hospital Southampton, contributed to recruitment and provided critical appraisal of the manuscript and approved the final version. MM is a trial coapplicant, contributed to trial set-up and recruitment and provided critical appraisal of the manuscript and approved the final version. HA contributed to the design and concept of the manuscript, provided critical appraisal of the manuscript and approved the final version. JPB is a principal investigator at University Hospital Southampton, trial coapplicant, provided critical appraisal of the manuscript and approved the final version. JPT is a chief investigator,provided critical appraisal of the manuscript and approved the final version.

**Funding** This project is funded by the Efficacy and Mechanism Evaluation (EME) Programme, an MRC and National Institute for Health Research (NIHR) partnership. EME reference 12/10/04. This paper presents independent research funded by the EME Programme and supported by the NIHR CRF and BRC at

Imperial College Healthcare NHS Trust and University Hospital Southampton NHS Foundation Trust.

**Disclaimer** The views expressed are those of the author(s) and not necessarily those of the EME Programme, the NHS, the NIHR or the Department of Health.

**Competing interests** AR received travel fees support from GI Dynamics. ADM has received honoraria for presentations and advisory board contribution by Novo Nordisk, Boehringer Ingelheim, AstraZeneca, Johnson & Johnson and research grant funding from Fractyl. AG reports funding supported by UK Medical Research Council and Wellcome Trust, outside of the submitted work. JPT received travel fees support from GI Dynamics. The rest of the authors report no conflicts of interest.

**Patient consent for publication** Not required.

**Provenance and peer review** Not commissioned; externally peer reviewed.

**ORCID iDs**
Aruchuna Ruban http://orcid.org/0000-0002-9105-4025
Hutan Ashrafian http://orcid.org/0000-0003-1668-0672

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
