## [Reviewer comments · BMJ Open]

ARTICLE DETAILS

TITLE (PROVISIONAL)	Effectiveness of different recruitment strategies in an RCT of a surgical device: Experience from the Endobarrier trial
AUTHORS	Ruban, Aruchuna; Pechtl, Christina; Glaysher, Michael; Chhina, Navpreet; Al-Najim, Werd; Miras, Alexander; Smith, Claire; Goldstone, Anthony; Patel, Mayank; Moore, Michael; Ashrafian, Hutan; Byrne, James; Teare, Julian

VERSION 1 – REVIEW

REVIEWER	Dr Li Wei Lead of Pharmacoepidemiology and Medication Safety Research Cluster UCL School of Pharmacy 29-39 Brunswick Square London, WC1N 1AX UK
REVIEW RETURNED	22-Jul-2019

GENERAL COMMENTS	This is a well written manuscript with clear information about their experience in recruiting patients into the Endobarrier trial. The authors may consider to address the following comments. 1. It is worth to provide all results in the abstract. Recruitment through GPs worked well in one site and in London Newspapers played a big role in recruiting patients.2. It is worth to calculate the costs per patient recruited by different strategy and this would provide useful information for others when they plan their studies.3. One of the reason for the difference in recruiting patients via newspapers between the SCOT trial and the recruitment in London could be due to the population density. The daily circulation of the newspapers are different between London and other places. Therefore it was not a surprise to see many calls generated from London when compared to the calls generated from Southampton.4. It is worth to explore the geographical variation further between the two sites such as relationship between GPs and patients, size of the GP surgery, life pace and life styles, etc. 4. Page 9, line 41: PIC Please provide full term for PIC when it appears first time 5. Page 21, line 42 Please correct the grammatical error
--

REVIEWER	Rob Andrews University of Exeter UK
-----------------	---

	I am a researcher who conducts studies in a similar area, doing surgical and lifestyle studies.
REVIEW RETURNED	07-Aug-2019

GENERAL COMMENTS	Review for BMJ open August 2019 Unfortunately, there are very few papers that report on recruitment to clinical studies which means that there is very little information out there about what works and what does not work. Thus, this paper by Dr Ruban and colleagues is needed by the research community, particularly as it is in area that there are few studies, surgical RCTs are rare. The paper is well written and easy to follow with only a few spelling mistakes. My comments are as follows Title Suggest this is changed to more accurately describe the paper. This paper gives insight into difficulties in recruit to one study and does not look at recruitment to a number of studies. Could be “Effectiveness of different recruitment strategies in a surgical/device RCT: Experience from the Endobarrier study” Abstract An abstract would normally summarise the finding of a paper. I think you have shown that the effectiveness of your recruitment strategies were different between London and Southampton. This needs to be highlighted. I think your conclusion is that for some areas extensive engagement with media may be needed and if so should be started early and come with appropriate support (although need to see data in more detail). Introduction The introduction is key as it needs to help the reader understand the context in which the study is being done and how this compares to other studies in this area. I thus think you need to cover  • What the disease area is being studied, frequency of the disease and difficulties in getting good HbA1c control. • Importance of recruiting to time – which you do • The difficulties in recruiting to studies ideally referencing diabetes studies – you do this but reference trials from many areas
---

- Details about the endobarrier studies – you have given some but there needs to be more – a bit of explanation about what endobarrier is and also an idea of what the patients have to do as part of the study. The study was quite intense with a lot of follow ups.

Methods

In this section please explain how you captured what source the patients were recruited from and how you confirmed the number of letters that were sent out by GPs. This is key as the response rate in London is 5% and in Southampton is 33% which is a 7 fold difference. That makes me worry that letters were not sent.

Please also reference the Southampton study that you used to base number of GPs on.

I was surprised that you expected recruitment to start as soon as you opened the study and for it to be completely linear. Normally there is a few months delay before recruitment and then an acceleration. Recruitment varies month to month and then tends to slow down at end.

Please note that

1. the weblink that you give for the websites do not all work.
2. In figure 2 there are words missing in the third box, the arrows are not straight and the number for the appendix is missing.

Results

The reader needs more detail here. What we need to know for each modality (and broken down by site) is

- Number who expressed interested
- Percentage of those who expressed interest who were not eligible (and ideally reasons why not eligible).
- Percentage of those expressed interest who were eligible
- Percentage of those who were eligible and were randomised

Having this information would enable to look at what is the most effective form of recruitment. It might well be that you had contact from 1004 people through advertisement but if only 20 came into the study then that is a lot of work for a small number of recruits. I would suggest making categories a bit larger.

For those that you know cost for you could also workout cost per patient recruited (advertising and GP).

Could you please check your recruitment flow diagram. I find it hard to believe that everyone who was eligible came into the study. Non eligible is anyone who meets your exclusion criteria. Eligible is anyone who meets your inclusion criteria. In both groups there will be people who cannot come into your study for other reasons such as Time, work pressure etc.

Please say how many GP practices in each area were used and how many letter sent out by GPs in each area.

Discussion

I would suggest that the discussion is more detailed and focused. Having the data mentioned above will help to compare and contrast the strategies in a more detailed way. I do not think that the dropout rate paragraph needs to be in. No data has been presented on this and this was not a focus of the paper. I am also unclear what the ENDO trail suspension section adds to the discussion and I would not have included this.

It would be nice to compare your recruitment and with other studies (if possible surgical or weight loss studies in the UK). For example

1. what is the recruitment rate per invitation letters in other studies? 5 % seems reasonable see <https://journals.plos.org/plosone/article/file?id=10.1371/journal.pone.0131521&type=printable>
2. On average how many people do you have to approach to get 1 person into a study?
3. On average how many GP practices would be needed to recruit 170 patients?

According to your figures your recruitment rate is 9% of people approached this is fairly similar to the hit rate I have had for the last 3 studies I have done but that is in type 1 population.

In terms of your conclusions – can not comment on whether they are correct until see the data mentioned above. It is though clear from your data that there will have to be flexibility in how recruitment is

	done for studies that recruit across a number of sites and that more knowledge about what works in different regions is important.
--	--

VERSION 1 – AUTHOR RESPONSE

Reply to Reviewers comments for BMJ Open Manuscript ID bmjopen-2019-032439

Section	Reviewer Comment	Authors Response
Title	Suggest this is changed to more accurately describe the paper. This paper gives insight into difficulties in recruit to one study and does not look at recruitment to a number of studies. Could be “Effectiveness of different recruitment strategies in a surgical/device RCT: Experience from the Endobarrier study”	Title has been changed accordingly to: ‘Effectiveness of different recruitment strategies to an RCT of a surgical device: Experience from the Endobarrier Trial’
Abstract	An abstract would normally summarise the finding of a paper. I think you have shown that the effectiveness of your recruitment strategies were different between London and Southampton. This needs to be highlighted. I think your conclusion is that for some areas extensive engagement with media may be needed and if so should be started early and come with appropriate support (although need to see data in more detail). It is worth to provide all results in the abstract. Recruitment through GPs worked well in one site and in London Newspapers played a big role in recruiting patients.	Abstract has now been amended accordingly to be more specific and to include more details of which modalities proved successful at the different sites respectively: “Despite these both being highly prevalent conditions, there were considerable barriers to the effectiveness of different recruitment strategies across each study site. Although recruitment from primary care proved extremely successful at one study site, this largely failed at another site prompting the implementation of multimodal recruitment strategies including a successful media campaign to ensure sufficient participants were enrolled and the study was adequately powered. From this experience we propose where appropriate the early engagement and investment in media campaigns to enhance recruitment into clinical trials.”

Introduction	The introduction is key as it needs to help the reader understand the context in which the study is being done and how this compares to other studies in this area. I thus think you need to cover  • What the disease area is being studied, frequency of the disease and difficulties in getting good HbA1c control. • Importance of recruiting to time – which you do • The difficulties in recruiting to studies ideally referencing diabetes studies – you do this but reference trials from many areas • Details about the endobarrier studies – you have given some but there needs to be more – a bit of explanation about what endobarrier is and also an idea of what the patients have to do as part of the study. The study was quite intense with a lot of follow ups. 	We do not feel that more background on obesity and diabetes and poor HbA1c control is required in the introduction for this manuscript as an extensive introduction was included as part of the protocol paper by the same research Group (Glaysher at al. 2017) which the reader is referenced to in this manuscript: 'The trial protocol including all details on the Endobarrier device and trial design has been previously published by our group, [5].' A recently published multi centre diabetes prevention trial has been referenced and added to the introduction section as an example of a diabetes study that took a long time to recruit with, low yield from screening visits[4].
Methods	In this section please explain how you captured what source the patients were recruited from and how you confirmed the number of letters that were sent out by GPs. This is key as the response rate in London is 5% and in	The text to explain how we confirmed number of letters that were sent out has now been included in the methods section: 'This process was reimbursed by the LCRN with £150 paid for the GP database search

	Southampton is 33% which is a 7 fold difference. That makes me worry that letters were not sent. Please also reference the Southampton study that you used to base number of GPs on.	and set up, £0.60 per participant information pack sent to patients, and £40 for each GP pre-screening questionnaire completed. Once a GP agreed to act as PIC for our trial, they initiated database searches to identify potential participants using two of the main inclusion criteria (BMI > 30 kg/m² and diagnosis of type two diabetes). The final number of eligible patients was then communicated back to the LCRN or research site who populated the adequate amount of patient packs (including patient information summary leaflet, recruitment invitation letter with response slip and prepaid envelope) and posted them back to the GP. The GP sent the packs out to each identified patient from their database. The same method was used across the two research sites, Imperial College Healthcare NHS Trust and University Hospital Southampton NHS Trust.' We now also explain how we captured the recruitment source: 'The clinical trials unit received regular updates on recruitment numbers and sources from each research site. This helped identifying recruitment challenges early and enabled the research teams to put new recruitment sources in place where necessary.'
Methods	I was surprised that you expected recruitment to start as soon as you opened the study and for it to be completely linear. Normally there is a few months delay before recruitment and then an acceleration. Recruitment varies month to month and then tends to slow down at end.	A sentence on recruitment predictions was added to the beginning of the discussion section: 'Despite a clear strategy from the offset and taking into account a non-linear recruitment rate with a delayed start at the beginning of the trial, recruitment took much longer than anticipated taking 2 years to complete rather than initially predicted 1 year.'

Methods	Please note that  1. the weblink that you give for the websites do not all work. 2. In figure 2 there are words missing in the third box, the arrows are not straight and the number for the appendix is missing. 3. Please provide full term for PIC when it appears first time 	1. The link was updated to: Study website –official websites for the trial were set up at each research site through the media office at Imperial College London and University of Southampton (https://www.imperial.nhs.uk/research/research-trials/diabetes-researchtrialswww.tinyurl.com/EB; https://www.southampton.ac.uk/medicine/academic_units/projects/endobarrier.page) and by the Imperial College research facility: http://imperial.crf.nihr.ac.uk/studies/endobarrier/http://imperial.crf.nihr.ac.uk/studies/EB/ 2. Figure 2 was updated as requested. This was a formatting issue which has now been
---------	--	---

		rectified. 3. The full term for PIC has been added.
Results	The reader needs more detail here. What we need to know for each modality (and broken down by site) is  • Number who expressed interested • Percentage of those who expressed interest who were not eligible (and ideally reasons why not eligible). • Percentage of those expressed interest who were eligible • Percentage of those who were eligible and were randomised • Having this information would enable to look at what is the most effective form of recruitment. It might well be that you had contact from 1004 people through advertisement but if only 20 came into the study then that is a lot of work for a small number of recruits. I would suggest making categories a bit larger. • For those that you know cost for you could also workout cost per patient recruited (advertising and GP). • It is worth to explore the geographical variation further between the two sites such as relationship between GPs and patients, size of the GP surgery, life pace and life styles, etc. • 	Some of the information requested by the Reviewer is available in Southampton but not London. In London, there was no study nurse available to support the research doctor with the sudden increase in recruitment and record keeping of these numbers. This was different in Southampton. The research doctor had support from the trials administrator and research nurses. Therefore a more detailed outline of the recruitment modalities and costs cannot be provided. Unfortunately, it is not possible to work out the average cost per patient for each recruitment modality as we would need to take the different advertisement campaigns into account that we run across sites. Also, because recruitment was delayed through GP PICs especially in London, we would potentially need to take the length of time into account to employ a research nurse or doctor to follow-up on this recruitment modality.

		This is nearly impossible to calculate or present these costing numbers.
Results	Could you please check your recruitment flow diagram. I find it hard to believe that everyone who was eligible came into the study. Non eligible is anyone who meets your exclusion criteria. Eligible is anyone who meets your inclusion criteria. In both groups there will be people who cannot come into your study for other reasons such as Time, work pressure etc. Please say how many GP practices in each area were used and how many letter sent out by GPs in each area.	More details in the text has been added to explain figure 4: “A flowchart summarising the overall recruitment figures from initial participant contact right through to randomisation is depicted in Figure 4. More details of the process of telephone screening, screening visit and randomisation can be found in the previously published protocol paper for the study,[5].” The recruitment flow chart has also been updated to make it more clearer.
Discussion	I would suggest that the discussion is more detailed and focused. Having the data mentioned above will help to compare and contrast the strategies in a more detailed way. I do not think that the dropout rate paragraph needs to be in. No data has been presented on this and this was not a focus of the paper.	We could not add a more detailed discussion on the recruitment modalities as the data was not available. The paragraph on dropout rate has been removed from the discussion.

Discussion	I am also unclear what the ENDO trial suspension section adds to the discussion and I would not have included this.	The failure of the ENDO trial in the US had a direct impact on recruitment for this study, leading to a substantial delay in recruitment as explained in this paragraph so we do feel it is vital that this is included in the manuscript.
Discussion	It would be nice to compare your recruitment and with other studies (if possible surgical or weight loss studies in the UK). For example:  1. What is the recruitment rate per invitation letters in other studies? 5 % seems reasonable see https://journals.plos.org/plosone/article/file?id=10.1371/journal.pone.0131521&type=printable 2. On average how many people do you have to approach to get 1 person into a study? 3. On average how many GP practices would be needed to recruit 170 patients? According to your figures your recruitment rate is 9% of people approached this is fairly similar to the hit rate I have had for the last 3 studies I have done but that is in type 1 population. 	A change to the text has been made under the discussion under section - Reflecting on the Success of Newspaper Advertising Campaign: ‘It must also be noted that although recruitment from GP practices was poor at the London site, the same was not observed at our Southampton site where recruitment from primary care was considerably better. This is in line with a trial that recruited participants for physical activity for individuals with diabetes which has been referenced,[12]. Researchers found that traditional recruitment approaches such as posting flyers and using clinical referrals was not successful whereby 77% of the participants were recruited using the electronic medical record system. This suggests that discrepancies in recruitment success in our trial could be site specific owing to the difference in patient populations between these two cities as previously identified which may have been a major contributory factor in these discrepancies in recruitment.’ However, as clinical trials usually do not present this level of recruitment detail, we were unable to comment on the recruitment rate per invitation letter in other studies.
Discussion		An additional explanation to a slower recruitment rates in London compared to Southampton has been added to the paragraph under - Participant information sheet (PIS) sub- optimal: ‘It also is important to note that there have been ethnical differences in the population that was approached at each research site with North West London representing a large Asian population and the area in and around Southampton representing a large white British population’

Conclusion	I cannot comment on whether they are correct until see the data mentioned above. It is though clear from your data that there will have to be flexibility in how recruitment is done for studies that recruit across a number of sites and that more knowledge about what works in different regions is important.	One sentence has been added to the conclusion: What is not so apparent is the best strategy to achieve this goal and so it is vital that there is flexibility in implementing variable recruitment modalities for multi-centre trials across different regions in England and the rest of the UK.
------------	--	--

VERSION 2 – REVIEW

REVIEWER	Li Wei University College London
REVIEW RETURNED	09-Oct-2019
GENERAL COMMENTS	A typo has not been corrected. Acknowledgments: Special thanks are given to Emanuela Falaschetti and Nicholas Johnson who were are dedicated statisticians for this clinical trial.